# 8q24.21 Locus: A Paradigm to Link Non-Coding RNAs, Genome Polymorphisms and Cancer

**DOI:** 10.3390/ijms22031094

**Published:** 2021-01-22

**Authors:** Claire Wilson, Aditi Kanhere

**Affiliations:** Department of Molecular Physiology and Cell Signalling, Institute of Systems, Molecular and Integrative Biology, University of Liverpool, Crown Street, Liverpool L69 3BX, UK; Claire.Wilson@liverpool.ac.uk

**Keywords:** long non-coding RNA, genetic polymorphisms, cancer

## Abstract

The majority of the human genome is comprised of non-protein-coding genes, but the relevance of non-coding RNAs in complex diseases has yet to be fully elucidated. One class of non-coding RNAs is long non-coding RNAs or lncRNAs, many of which have been identified to play a range of roles in transcription and translation. While the clinical importance of the majority of lncRNAs have yet to be identified, it is puzzling that a large number of disease-associated genetic variations are seen in lncRNA genes. The 8q24.21 locus is rich in lncRNAs and very few protein-coding genes are located in this region. Interestingly, the 8q24.21 region is also a hot spot for genetic variants associated with an increased risk of cancer. Research focusing on the lncRNAs in this area of the genome has indicated clinical relevance of lncRNAs in different cancers. In this review, we summarise the lncRNAs in the 8q24.21 region with respect to their role in cancer and discuss the potential impact of cancer-associated genetic polymorphisms on the function of lncRNAs in initiation and progression of cancer.

## 1. Introduction

Until recently, proteins and DNA were considered the main functional molecules of cells, with RNA being only a messenger between the two. This is the main reason why it was quite surprising to find that merely 2% of the human genome codes for proteins while 98% is non-protein coding. Since the human genome was first sequenced in 2003 [1], the biggest puzzle has been to understand the necessity of the 98% non-coding genome. Recent advances in sequencing technologies, however, have revealed that a large majority of the non-coding genome is transcribed into RNA transcripts. These non-coding RNAs, or ncRNAs, if functional, can explain the mystery of the non-coding human genome. However, functions of a large majority of the ncRNAs remain unknown, raising the possibility that they are merely a result of transcriptional noise in the cell. On the other hand, a number of observations support the functional importance of non-coding RNAs. For example, the observation that nearly 90% of all disease-associated genomic variations are within non-coding transcripts points towards their functional role [2]. However, a detailed analysis of genetic variations in the ncRNAs and their significance in diseases is needed. In this review, we will discuss disease-associated single-nucleotide variations in long non-coding RNAs or lncRNAs that are expressed from the 8q24.21 locus and their implications in cancers.

## 2. LncRNAs and Their Functions

LncRNAs are non-coding RNA transcripts that are more than 200 nucleotides in length [3]. With over 50,000 lncRNAs currently annotated and more still being discovered, the number of lncRNAs is significantly higher than that of protein-coding genes in the human genome [4]. LncRNAs are transcribed from stretches of DNA sequence located between genes (intergenic lncRNAs) or from introns of protein-coding genes (intronic lncRNAs). Much of the current research on lncRNAs has focused on intergenic lncRNAs or lincRNAs due to the difficulty of deciphering the function of intronic lncRNAs from the protein-coding gene that they reside in [5].

Although the functionality of a large majority of lncRNAs still remains unknown, examples suggest that lncRNAs can participate in diverse functions in the nucleus and cytoplasm, playing roles from transcription to translation (Figure 1). Often lncRNAs are located within the nucleus, where they are involved in chromatin remodelling, and transcriptional and post-transcriptional regulation [6]. Some lncRNAs are also exported to the cytoplasm. Within the cytoplasm, lncRNAs regulate translation, mRNA turnover and post-transcriptional modifications [6]. Some of the well-characterised lncRNAs, such as the X chromosome inactivation regulator *Xist*, can in fact co-ordinate multiple functions [7].

Within the nucleus, lncRNAs can act to regulate transcription through binding and sequestering proteins to prevent them from binding to their target RNA [8]. One set of proteins which can be modulated by lncRNAs are transcription factors. The lncRNA *linc-YY1* has been shown to act as an RNA decoy for the transcription factor yin yang 1 or YY1, preventing the protein from binding to target promoters [9]. In addition, lncRNAs have been shown to bind to chromatin modifiers to alter transcription. Depending on the chromatin complex, lncRNAs can act to either repress or activate genes. For example, *Kcnq1ot1* has been shown to repress transcription through binding and recruiting the DNA methyltransferase DNMT1, resulting in increased DNA methylation [10]. Contrastingly, the lncRNA *HOTTIP* binds to an adaptor protein to promote histone H3 lysine 4 trimethylation and gene transcription [11].

LncRNAs can also act as protein scaffolds and form ribonucleoproteins. In the nucleus, these complexes can regulate gene transcription. For example, *lincRNA-p21* recruits the heterogeneous nuclear ribonucleoprotein-K or hnRNP-K to the promoter region of p21 and activates transcription [12]. In the cytoplasm, interactions between lncRNAs and ribonucleoproteins can regulate translation. For example, *lincRoR* interacts with hnRNP-I and supresses p53 translation by preventing the hnRNP-I from interacting with p53 mRNA [13].

In the cytoplasm, lncRNAs can modify gene expression post-transcriptionally through various mechanisms. LncRNAs can bind to microRNAs or miRNAs through complementary base pairings. This prevents miRNA from binding to target mRNAs, therefore enhancing expression of these genes by increasing translation and/or stability of mRNA. For example, the lncRNA *SNHG7* has been identified to sponge miR-216 resulting in increased expression of the miR-216 target gene *GALNT1* [14]. LncRNAs can also mediate mRNA decay though interacting with RNA-binding proteins. An example is a group of identified lncRNAs called half(½)-sbsRNAs which can bind to mRNA and form a functional Staufen 1-binding site, therefore triggering Staufen 1-mediated mRNA decay [15]. To regulate genes post-transcriptionally, lncRNAs can influence splicing events. *MALAT1* has been shown to influence mRNA splicing by interacting with serine/arginine splicing factors and influencing the distribution of these splicing factors [16]. Finally, lncRNAs can act as translation inhibitors through binding to proteins associated with translation regulation. For example, *lincRNA-p21* inhibits translation of target mRNAs through the recruitment of the translation repressor Rck [17], while *AS Uchl1* acts as a translation activator of Uchl1 via an embedded SINEB2 element [18].

## 3. LncRNAs, GWAS and Cancer

Cancer is often described as a genetic disease arising from dysfunction of genes which are responsible for normal cell division and growth. Somatic mutations during one’s lifetime are believed to underlie the development of most cancers [20]. In a proportion of cancers, hereditary mutations confer a greater risk for the disease. In these cases, while some high-penetrance cancer susceptibility genes have been identified, such as *BRCA1* for breast cancer, it is believed that genetic susceptibility in the majority of cases is a result of combined effects of common low-penetrance alleles and rare disease-causing variants [21]. Genome-wide association studies or GWAS have helped to identify novel genes associated with cancer, which has in turn helped improve not only our understanding of the disease but also drive forward new therapeutic advances related to population screening and pharmacological approaches.

Most causal variants associated with cancer risk identified through GWAS are located within non-coding regions of the genome [21]. Since the identification of lncRNAs, emerging evidence has begun to suggest that lncRNAs play a pivotal role in cancer development. Oncogenes are mutated genes which drive cancer initiation and progress. Under normal conditions, these genes are termed protooncogenes and are responsible for cell division and growth [22]. Like protooncogenes, lncRNAs have been proposed to regulate key cellular mechanisms associated with tumourgenesis including cell proliferation, apoptosis, migration and invasion [23]. For example, the lncRNA *HOTAIR* has been shown to promote proliferation and invasion of cancer cells in part through the regulation of p53 transcriptionally and *SOX2* post-transcriptionally [24]. LncRNAs also have a role in epigenetic regulation including DNA methylation and histone modification which are important processes in cancer initiation [25]. In this role, *HOTAIR* can recruit histone-modifiers and chromatin remodelling complexes to regulate gene transcription [26]. In breast cancer, the interaction of *HOTAIR* with the Polycomb Repressive Complex 2 results in changes in histone methylation and gene expression associated with an increase in cancer cell metastasis [27]. *HOTAIR* highlights how a lncRNA can have a multi-faceted role in cancer development.

Investigating the role of specific lncRNA in cancer phenotypes is therefore a popular area of research. The identification of causal variants within non-coding loci has helped highlight potentially important lncRNAs related to cancer development.

## 4. 8q24.21

The human genome locus, 8q24.21, has been previously described as a “gene desert” due to sparse presence of protein-coding genes in this 4.1 MB region [28] (Figure 2). Despite this, genetic variation within this area has been associated with numerous cancer phenotypes. Lying within this region is one of the most studied oncogenes *MYC*, which is estimated to be involved in 20% of human cancers [29]. *MYC* primarily acts as a transcription factor and is shown to regulate cell cycle, metabolism, ribosome biogenesis and cell adhesion [29]. Numerous genetic polymorphisms associated with cancer risk are located within the non-coding regions surrounding *MYC*, which has led to characterising these regions as regulatory elements influencing *MYC* expression [30]. Other protein-coding genes within this region have also been shown to contribute to tumourigenesis [31,32,33,34]. These are *FAM84B*, *GSDMC*, *FAM49B*, and *ASAP1*. In addition, the *OCT4* pseudogene *POU5F1B* is also present in the 8q24.21 region and has been showed to be amplified in cancer [35]. While *POU51B* has been referred to as a pseudogene, the latest human genome assembly GRCh38 from the Genome Reference Consortium states that *POU5F1B* is protein coding. However, in addition to the oncogenic protein-coding genes, the region 8q24.21 is a host to numerous lncRNAs associated with various cancers and with functions that are independent of *MYC*.

A total of 12 lncRNAs are located within the 8q24.21 region (Figure 2). Upstream of the protein-coding gene *MYC* are the lncRNAs *PCAT1*, *CASC19*, *PRNCR1*, *CCAT1*, *CASC8*, *CCAT2* and *CASC11*. Downstream of *MYC* lies *PVT1*, *LINC00924*, and *CCDC26*. The majority of the lncRNAs located within 8q24.21 can be categorised as lincRNAs as they are located in intergenic regions between the five protein-coding genes in this region. The exception is *CASC8*, which is anti-sense to the protein-coding gene *POU5F1B.* In addition to lncRNAs, there are also several microRNAs or miRNAs located within the region. Five miRNAs have been shown to arise from the *PVT1* locus. These are miR-1204, miR-1205, miR-1206, miR-1207 and miR-1208. In addition, miR-3686 is located within *CCDC26*, with miR-5194 located downstream of *CCDC26*.

Numerous genetic variants associated with increased risk of cancer are located within the lncRNAs in the 8q24.21 locus. According to the NHGRI-EBI GWAS Catalog [36], a total of 324 single-nucleotide polymorphisms or SNPs have been reported in the 8q24.21 region. Interestingly, to date, only four SNPs have been located within the *MYC* locus and none of these SNPs are associated with cancer. Contrastingly, several of the lincRNAs in this region appear to be hotspots for SNPs including *PCAT1* and *CCDC26* which are associated with 73 and 81 SNPs, respectively, with many of these SNPs associated with various cancers. While it is possible that SNPs located in the non-coding regions may influence protein-coding genes, they also highlight the potential importance of the 8q24.21 lncRNAs in disease [37]. Therefore, unsurprisingly, many studies have focused on the 8q24.21 region to help elucidate the role of lncRNAs in the initiation and progression of cancer in the hope of identifying novel biomarkers and new therapeutic targets. In this review, as a paradigm, we summarise the potential impact of genetic polymorphisms on tumourigenic functions 8q24.21 lncRNAs.

### 4.1. 8q24.21 LncRNA Expression Changes in Cancer

Relative to protein-coding RNA, the expression levels of lncRNA are low [38]. However, lncRNAs are often found to be elevated cancer and contribute to the transition of normal cells into tumour cells [39]. The lncRNAs located within the 8q24.21 locus have been shown to be upregulated in a number of cancers including colon cancer, pancreatic cancer, glioma and leukaemia (Table 1). While the functional significance of the increased expression is not fully known, upregulation of these lncRNAs often correlated with poorer overall survival and a more aggressive cancer phenotype. Therefore, high expression of the lncRNAs appears to be an independent prognosis factor for advanced cancer and poorer prognosis in various cancer subtypes. The impact of the lncRNAs in promoting a tumourigenic phenotype is attributed to their ability to drive proliferation, migration and invasion of cancer cells. Silencing of these lncRNA genes was found to supress cell growth and apoptosis in cancer cells. Targeting these lncRNAs therefore appears to be a promising therapeutic approach for many cancers.

### 4.2. 8q24.21 LncRNAs in miRNA Regulation

Numerous binding sites of miRNAs have been located within the lincRNAs of the 8q24.21 locus. MiRNAs are small ncRNAs which are an average of 22 nucleotides in length. The main function of miRNAs is in gene silencing, in which they bind to mRNAs, predominantly the 3′UTR region, and supress expression [117]. MiRNAs have been shown to regulate tumourigenesis through the suppression of oncogenes and tumour suppressor genes, and dysregulation of miRNAs is evident in various cancers [118]. LncRNAs have been shown to act as a molecular sponge to sequester miRNAs and prevent their interaction with target messenger RNAs. Studies have identified complementary base pair regions between the majority of the lincRNAs located in 8q24.21 (*PVT1* [119], *CCAT1* [120], *PCAT1* [121], *CCAT2* [122], *CASC11* [123], *PRNCR1* [52], *CASC19* [50], *CCDC26* [116]) and miRNAs which have been previously identified as tumour suppressors. Within this role, many of these lincRNAs can act as competitive endogenous RNA or ceRNA for miRNA target genes (Table 2). Like other miRNA target genes, lncRNA expression changes are often associated with the initiation and/or progression of cancer. A good example of this is the 8q24.21 lincRNA *CCAT2*, which was found to act as a ceRNA, in glioma, for miR-424, which also targets VEGFA. In this case, subsequent activation of VEGFA signalling promotes angiogenesis [89].

### 4.3. 8q24.21 LncRNAs and Chromatin Modifications

LncRNAs can often modulate gene expression through changes in chromatin modifications such as DNA and histone methylation. One such example is that of lincRNA *CCDC26*, which is expressed from a locus approximately 1MB downstream of *MYC*. Within numerous cancer lines, *CCDC26* has been shown to regulate cell growth and apoptosis [116,143,144], and be enriched within nuclear fraction of myeloid leukaemia cells [144]. We have further explored the role of *CCDC26* in apoptosis in relation to DNA methylation. We found that *CCDC26* interacts with the methyltransferase DNMT1 and drives changes in methylation levels in genomic DNA [144]. Inhibition of *CCDC26* results in the hypomethylation of the genome, leading to inhibition of cell growth and an increase in apoptosis associated with DNA damage. While DNMT expression levels remained unchanged following *CCDC26* removal, DNTM1 was found to be mis-localised in cytoplasm which is the likely cause for the observed genomic hypomethylation [144]. Our results suggest that *CCDC26* influenced subcellular localisation of DNMT1 through protein–RNA interaction in the nucleus [144].

Overexpression of another lincRNA *PVT1* in 8q24.21 resulted in increased levels of the methyltransferases DNMT1, DNMT3a and DNMT3b and promoted the methylation CpG islands located in the miRNA miR-146a promoter region [145]. The methylation of the promoter region resulted in the suppression of miR-146a expression. While the role of miR-146a in cancer was not studied further, significantly low expression of miR-146a was found in prostate cancer cells compared to normal tissue, suggesting a role in tumour development [145]. In addition, lincRNAs *PVT1* and *PCAT1* have been shown to directly bind to the histone methyltransferase EZH2 and induce trimethylation of lysine 27 on histone H3, H3K27me3, which leads to transcriptional repression. *PVT1* induces H3K27me3 modification at the miR-200b promoter, leading to silencing of miR-200b [146]. Mir-200b has been shown to inhibit proliferation and migration of cancer cells, and is dysregulated in numerous cancers [146]. Similarly binding of EZH2 to *PCAT1* is needed for *PCAT1*-associated silencing of E-cadherin and p21 [147,148]. LincRNA *CCAT2* has also been shown to bind directly to EZH2 as well as post-transcriptionally downregulate EZH2 expression [149].

### 4.4. 8q24.21 LncRNAs and c-Myc Regulation

Many studies have investigated the link between the lncRNAs located on 8q24.21 and the oncogene *MYC* and the protein it encodes c-Myc, especially lncRNAs *PVT1* and *PCAT1*, which lie in a close proximity to *MYC*. Co-amplification of *PVT1* and *MYC* is evident in numerous cancers [150]. Studies have suggested that *PVT1* and *MYC* work synergistically in the regulation of proliferation [151]. Indeed, many of *PVT1* targets are also downstream signalling molecules of c-Myc, strengthening the idea that they both work together to drive cancer progression [151]. *PVT1* is involved in the transcriptional and post-transcriptional regulation of c-Myc [151,152,153]. In addition, *PVT1* itself is a target for c-Myc. Within the *PVT1* promoter region, two enhancer E-box sites have been identified in which c-Myc binds to transcriptionally activate *PVT1* [154]. Therefore, it is possible that a positive feedback mechanism exists between *PVT1* and c-Myc, in which c-Myc promotes *PVT1* expression and then *PVT1* prevents the degradation of c-Myc, driving the expression of both oncogenes in cancer.

Contrastingly, a study has found that silencing the *PVT1* promoter results in increased cell proliferation and competition in breast cancer cells, which was associated with an increase in *MYC* expression [155]. The study identified that four intragenic enhanced elements have been identified in the *PVT1* gene which bind to the *PVT1* promoter in a preferential manner over *MYC*. Silencing of the *PVT1* promoter resulted in these enhancer elements making stronger connections with the *MYC* promoter and enhancer region resulting in increased *MYC* expression. Altogether, the study proposed that the *PVT1* promoter can act to repress *MYC* transcription in a *cis* manner [155].

On a post-transcriptional level, *PCAT1* has also been found to regulate c-Myc. Overexpression of *PCAT1* resulted in the post-transcriptional upregulation of c-Myc [156]. *PCAT1* was also shown to interfere with miRNA-mediated degradation of *MYC*. The upregulation of c-Myc by *PCAT1* promotes cell proliferation and the expression of genes involved in protein biosynthesis and transcriptional elongation [156]. Supporting this, the knockdown of *PCAT1* also resulted in reduced cell proliferation associated with downregulation of the c-Myc/MAPK signalling pathway [45].

In addition to *PVT1* and *PCAT1*, other lncRNAs from the 8q24.21 locus are shown to regulate *MYC* expression. LincRNA *CASC11*, for example, transcriptionally regulates *MYC* expression by modulating a *MYC* enhancer that is 557 kb upstream of *MYC*. The binding occurred in a PCBP2-dependent manner, with *CASC11* and the RNA-binding protein PCBP2 forming a looping structure between the *MYC* promoter and enhancer [157]. Similarly, lincRNA *CCAT1* has been shown to be required for the maintenance of chromatin loops at the *MYC* locus in colorectal cancer. The study suggested that *CCAT1* interacts with CTCF, a transcriptional regulator protein, and regulates the binding of CTCF to chromatin, which mediates the long-range interactions between *MYC* promoter and enhancers [158].

Therefore, targeting lncRNAs which modulate c-Myc activity proposes a novel approach to treating Myc-associated cancers.

## 5. Potential Impact of Genetic Polymorphisms on 8q24.21 LncRNAs

### 5.1. Amplifications

Amplifications within the 8q24.21 region have been detected in various forms of cancer. This genomic site is known as one of the most prevalent sites of copy number gains in cancer [159,160]. The presence of copy number gains in this region could result in upregulation of lncRNAs and promote abnormal cell growth of cells (Figure 3A). In a study focusing on ovarian cancer, gains localised to 8q24.21 were evident in 59% of samples and resulted in a significant overexpression of *PVT1*, suggesting gains within this region can be a crucial factor of elevating lncRNAs in cancer [161,162]. In addition, copy number variants have been shown to drive co-amplification of *PVT1* and *MYC* in numerous cancers [152]. Analysis of The Cancer Genome Atlas database found that out of the tumours which displayed an 8q24 copy-number increase, 97% showed a gain of both *PVT1* and *MYC*, while less than 1% of tumours showed an increase in copy number of *MYC* alone [152]. This gain of both *PVT1* and *MYC* has been shown to be essential for tumourigenesis in 8q24-amplified cancer cells. In the study, gain of *MYC* or the *PVT1*/*CCDC26*/*GSDMC* region alone did not result in tumourigenesis, but co-amplification of *MYC* and *PVT1*/*CCDC26*/*GSDMC* resulted in a pro-tumour transformation. Specifically, the *PVT1* copy number increase was found to be the critical factor for *MYC*, and subsequent c-Myc, elevation in *MYC*-driven cancers [152].

In addition to *PVT1*, amplifications in other 8q24.21 lncRNAs have been identified. In studies investigating paediatric acute myeloid leukaemia, the most common copy number alteration was found to be a low burden increase in the *CCDC26* locus [163]. In castration-resistant prostate cancer samples, duplications hotspots were evident in the region hosting *PCAT1* and *PRNCR1* [164]. While the study did not investigate expression of the lncRNAs, it is possible that these events lead to amplification of these genes.

### 5.2. Structural Variations

Chromosomal instability is a source of genetic variation and is a common feature of tumours. Often this leads to chromosomal rearrangements including translocations [165]. Within the 8q24.21 loci, a common site of translocation is within *PVT1*. *PVT1* was initially identified as a site for variant translocation involving the Ig kappa locus and *PVT1* gene in murine plasmocytomas [166], with the equivalent translocation observed in Burkitt’s lymphoma in human [167]. Since then, recurrent intra-chromosomal inversions within intron 1 of *PVT1* have been identified in breast cancer patients [155]. Similar to breast cancer, formation of *PVT1* fusion proteins as a result of translocations occurring in intron 1 of the gene is also evident in numerous other cancers (Figure 3B) [155,168,169]. The promoter region of *PVT1* appears to be critical to the function of the lincRNA, especially in relation to c-Myc regulation. The intra-chromosomal inversion, in addition to forming a fusion protein, is also thought to affect the *PVT1* promoter region by separating it from intragenic enhancers [155]. *PVT1* has also been identified to fuse with *MYC* [150], *CCDC26* [161], *NSMCE2* [170], and *NDRG1* [171]. While the specific effects of these gene fusions are unclear, it is possible that they have functions distinct from the individual genes. For example, in a breast cancer model, gain of both *PVT1* and *MYC* resulted in an increased proliferation of cells which was not evident by increase in *PVT1* and *MYC* alone [172].

Other genomic rearrangements which often occur alongside amplification events include the formation of double-minute chromosomes. These extrachromosomal fragments of DNA consist of repeats of specific chromosomal regions termed an amplicon. Amplification of double minutes are believed to increase the copy number of oncogenes to drive tumour heterogeneity and are associated with treatment resistance [173]. For example, in acute myeloid leukaemia patients, 8q24 chromosome is often amplified by forming double-minute chromosomes [174,175]. Surprisingly, evidence suggests that *MYC* is not the target gene for this type of amplification due to the gene often being silent in these subsets of acute myeloid leukaemia cases [174]. Therefore, it is possible that the target of the amplicon is the lncRNAs residing within this region. Supporting this idea, mapping of the 8q24.21 amplicon in acute myeloid leukaemia has identified alterations in the structure of lincRNA *CCDC26* as a result of recombination event upstream of exon 4 [175]. It is unknown whether this partial amplification of the gene results in a gain or loss of function, but it is likely abnormalities in *CCDC26* structure drive a pro-tumour phenotype.

### 5.3. Single-Nucleotide Polymorphisms

Despite a large number of cancer-associated SNPs being located within lncRNAs in the 8q24.21 locus, studies have yet to identify the impact of these genetic polymorphisms on the functions of the lncRNAs. One SNP that has been explored is rs698267, located within *CCAT2*, and it is highly prevalent in patients with myelodysplastic syndrome [149]. The *CCAT2* SNP was shown to result in a specific RNA editing event causing *CCAT2* DNA-to-RNA allelic imbalance, with the RNA transcribed from the SNP locus being different compared to corresponding genomic DNA. The cancer-predisposing allele resulting from this SNP resulted in gene expression dysregulation and repression of EZH2 function [149]. Interestingly, of the patients carrying the SNP, 78% expressed heterozygous *CCAT2* RNA. The study found that the presence of both the normal and SNP-carrying *CCAT2* allele resulted in a combined oncogenic effect and a stronger downregulation of EZH2 than that achieved by homozygous expression of either allele [149]. This phenomenon is believed to be due to the specific alleles independently promoting a pro-tumour phenotype through regulating distinct pathways [149]. The same SNP has also been investigated in relation to cancer metabolism [176]. In this study, the SNP was shown to induce changes in the secondary structure of *CCAT2* and affect the binding of *CCAT2* to the CFIam protein. This resulted in alternative splicing of *GLS* to preferentially induce the expression of the oncogenic isoform [176].

The impact of SNPs on 8q24.21 lncRNAs can be guessed based on the position of the SNPs within the lncRNA genes. Studies investigating lncRNA SNPs in other areas of the genome have shown that SNPs can influence the expression of lncRNAs (Figure 3A). A cancer-associated SNP located with the lncRNA *PTCSC3* reduces the binding affinity of the transcription factor C/EBPα, resulting in decreased *PTCSC3* promoter activation and reduced expression of the lncRNA which is evident in thyroid carcinoma [177]. Within the 8q24.21 region, a large number of cancer-associated SNPs are clustered within the first intron of *CCDC26* gene. On the background that the main transcriptional start site of *CCDC26* is located within exon 2 of the gene [144], we can speculate that cancer-associated SNP hotspot in the first intron may affect the promoter region of *CCDC26* and therefore impact expression of the lincRNA. Similarly, SNPs associated with cancers have been shown to be located within intron 1 of *PVT1* which can modify promoter strength of the gene [36].

In addition, a cancer-associated SNP located within the intronic region of the lncRNA *PCAT19* has been found to regulate the isoforms of *PCAT19* in a reciprocal manner. The SNP is associated with decreased levels of *PCAT1*-short and increased levels of *PCAT1*-long, with the long isoform associated with prostate cancer progression [178]. Many of the lncRNAs located in the 8q24.21 region have numerous different isoforms. While the functional role of the varied isoforms has not been assessed, it is possible that SNPs within the intronic regions of these lncRNAs could mediate a switch between isoforms in cancer (Figure 3C).

SNPs can also impact the structure of lncRNAs and their ability to form RNA and protein interactions. Studies have begun to look at this concept by using computational modelling to predict the impact of SNPs on lncRNA function. A lung cancer-associated SNP located in exon 2 of *NEXN*-1 has been predicted to change the secondary structure of the lncRNA, which may influence the ability of *NEXN*-1 to interact with proteins [179]. In addition, a SNP located within the intronic region of *HOTAIR* has been shown to alter the binding affinity of the lncRNA to transcription factors associated with cancer including *PAX-4, SOX, SPZ1* and *ZFP281* [180]. In terms of 8q24.21 lncRNAs, no studies have yet investigated the impact of SNPs on their ability to bind to target protein. However, it is probable that SNPs influence the function of 8q24.21 lncRNA in gene regulation, notably in relation to *MYC* (Figure 3D).

Cancer-associated SNPs have also been identified within miRNA binding sites of lncRNAs [181]. These SNPs can interfere or enhance hybridisation of the lncRNAs with miRNAs. A SNP located in the lncRNA *MALAT1* has been shown to inhibit binding of the lncRNA to miR-194-5p, resulting in increased expression of *MALAT1* and contributing to colorectal cancer development [182]. SNPs can also result in new binding sites for miRNA. A SNP identified in *CCSlnc362* which confers a protective role from colorectal cancer was shown to result in a new miRNA binding site for miR-4658 in exon 1 of the lncRNA. Binding of miR-4658 reduced expression and supressed the role of *CClnc362* in tumour progression [183]. While these studies have focused on the role of miRNAs on the expression of lncRNAs, it is possible that SNPs within miRNA binding domains can impact the ability of lncRNAs to sponge miRNAs, a feature common to 8q24.21 lncRNAs. In this role, SNPs could result in novel lncRNA-mediated miRNA sponging events or prevent lncRNA:miRNA binding, resulting in increased expression of cancer-associated miRNAs (Figure 3E).

In complex diseases such as cancer, the underlying genetic cause is usually a result of multiple genetic polymorphisms. SNPs have been shown to co-occur with other SNPs, and it is likely that this occurs in the 8q24.21 region. While individual SNPs generally confer a small effect, the combined effect of numerous relevant SNPs is thought to result in a synergistic increased risk. In addition, co-occurrence of SNPs can lead to SNP–SNP interactions [184]. A study investigating pancreatic cancer risk identified numerous SNP–SNP interactions involving SNPs located in the 8q24.21 region. Interestingly, a SNP located in *CASC11* was involved in 70% of the SNP–SNP interaction pairs significantly associated with pancreatic cancer risk, with the most common genetic interaction being between *PVT1* and *CASC11* [185]. Although a direct link between the two lncRNAs still remains missing, it is thought that *PVT1* and *CASC11* are linked through *MYC* [91,185,186].

## 6. Future Work and Limitations

Genetic polymorphisms are a frequent feature of cancer. While not necessarily of greater importance compared to other cancer polymorphisms, 8q24.21 can serve as an excellent paradigm for understanding the role of lncRNAs and non-coding genetic polymorphisms in oncogenesis. A possible approach to assessing the importance of 8q24.21 lncRNAs is through the use of animal models. However, there are limitations. Firstly, the majority of the lncRNAs in the 8q24.21 locus are not evolutionary conserved, and there are no mouse orthologs for these lncRNAs. Despite this, some parts of the non-coding region of 8q24.21 have been evaluated in mouse models [187]. One study generated a mouse model lacking a 430 kb gene desert region between the protein-coding genes *FAM84B* and *MYC*. This region had previously been associated with breast cancer susceptibility in humans. The study found that deletion of this region resulted in an anti-tumourigenic effect in early and late stages of mammary cancer in mouse breast cancer models [187]. Interestingly, deletion of the region resulted in a decreased expression of *MYC*, located approximately 200 kb from the deleted region. The study further identified a chromatin loop conserved between mouse and human from the *MYC* promoter to the deleted region [187]. These findings suggest that the 8q24.21 gene desert region does contain regulatory elements which are conserved in both humans and mice. The evolutionary relation of these conserved regions with lncRNAs found in the human 8q24.21 region needs to be scrutinised.

While animal models have been effective in the study of some evolutionary conserved lncRNAs, some discrepancies between human and animal data have raised questions regarding functional significance of lncRNAs. For example, the evolutionary conserved, cancer-associated lncRNA *MALAT1* has been suggested to regulate gene transcription in human cells. However, mouse models lacking *MALAT1* have shown no obvious phenotypes and had no apparent changes in global gene expression or pre-mRNA splicing [188]. Altered expression of *MALAT1* has been described in many human cancers, with in vitro studies indicating that *MALAT1* promotes tumour growth. Studies investigating reduced *MALAT1* expression in mouse models of cancer have found conflicting results depending on the strategy of *MALAT1* inactivation [188]. Using the MMTV-PyMT (mouse mammary tumour virus-polyoma middle tumour antigen) cancer model, knockout of a 3 kb portion of *MALAT1* promoter region and the lncRNAs 5′ end resulted in a reduction in lung metastases. On the other hand, using the same MMTV-PyMT model but with targeted insertional inactivation of *MALAT1* resulted in an increase in metastatic sites in the lungs [188]. The work involving *MALAT1* highlights the limitations in current understanding and the need for robust characterisation of lncRNA evolution and function.

## 7. Conclusions

Genetic polymorphisms associated with cancers are often located within the non-coding region of the genome. The 8q24.21 locus is the site of numerous lncRNAs and a hotspot of cancer polymorphisms. Studies have identified elevated expression of the 8q24.21 lncRNAs in numerous cancers associated with the progression of tumour phenotype. Indeed, in various studies, high levels of these lncRNAs have been identified as an independent prognostic factor for poor overall survival. The potential importance of these lncRNAs is evident by functional studies showing that, within cancer tissues and cells, the 8q24.21 lncRNAs sponge miRNAs drive chromatin modification and regulate the expression of the oncoprotein c-Myc. The 8q24.21 region of the genome is therefore an interesting region to explore the link between genetic variants, lncRNAs and cancer progression. While the advent in GWAS has highlighted the large number of cancer-associated variants within this region, research is still lacking on identifying the impact that these polymorphisms have on lncRNA function. We can begin to speculate the potential impact of genetic polymorphisms, including SNPs, based on previous studies investigating the impact of variants on lncRNA function. Further work to help bridge this knowledge gap is needed which will allow a clearer understanding of the functionality of lncRNAs in cancer and will boost the use of lncRNAs as novel targets in cancer therapeutics.

## Figures and Tables

**Figure 1 ijms-22-01094-f001:**
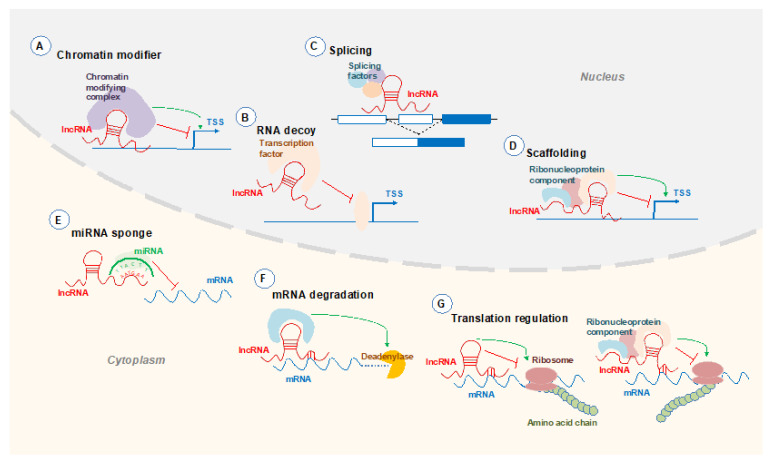
Mechanisms of lncRNAs. LncRNAs (in red) function in the nucleus and cytoplasm to act on transcriptional, post-transcriptional and translational levels. Within the nucleus, lncRNAs can regulate transcription through (**A**) recruiting chromatin modifying complexes to supress or activate transcriptional start sites (TSS) or (**B**) modulating transcription factor activity. LncRNAs can also (**C**) interact with mRNA to alter splicing events and (**D**) act as a scaffold and bind numerous proteins to form a ribonucleoprotein complex which can regulate transcription within the nucleus and (**G**) translation in the cytoplasm. In the cytoplasm, lncRNAs can also regulate gene expression post-transcriptionally by (**E**) acting as a microRNA (miRNA) sponge or interact with mRNA to (**F**) promote degradation or (**G**) regulate translation [19].

**Figure 2 ijms-22-01094-f002:**
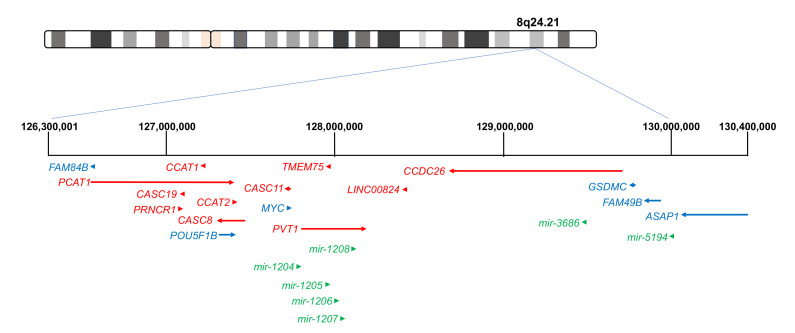
Map of the 8q24.21 region of the human genome. The 8q24.21 region (chr8:126,300,001–130,400,000) is predominately a non-coding region containing numerous lncRNAs (in red), with some microRNAs (in green) and few protein-coding genes (in blue). The direction of arrows indicates the direction in which the genes are transcribed.

**Figure 3 ijms-22-01094-f003:**
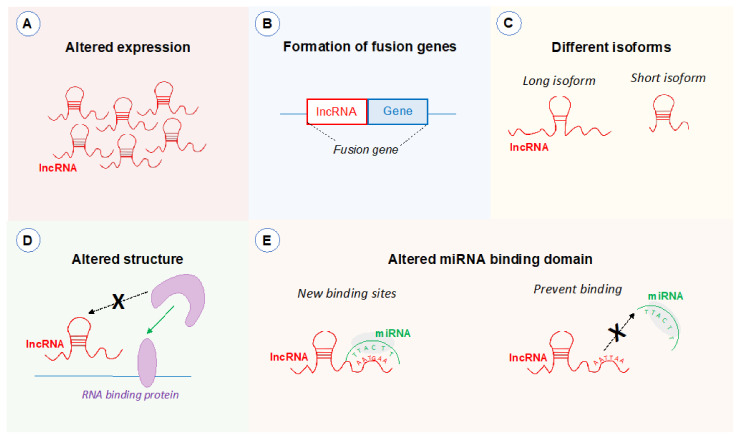
Potential effects of genetic variants on 8q24.21 lncRNA expression and function. (**A**) Copy number gains or single-nucleotide polymorphisms (SNPs) within the promoter region can alter the expression of lncRNAs (in red). (**B**) Chromosomal translocations can lead to the formation fusion genes containing 8q24.21 lncRNAs. (**C**) An intronic SNP can result in different lncRNA isoforms being expression including short and long versions of the lncRNA. SNPs located in intronic or exonic regions can result in changes in the structure, therefore (**D**) preventing binding to proteins, or (**E**) alterations in the miRNA binding domains resulting in either new binding sites or preventing binding to target miRNAs.

**Table 1 ijms-22-01094-t001:** Summary of cancers associated with lncRNAs located in the 8q24.21 region.

8q24.21 LncRNA	Cancers in Which the lncRNA Has Been Studied
*PCAT1*	Colorectal cancer [40], hepatocellular carcinoma [41], extrahepatic cholangiocarcinoma [42], oesophageal squamous carcinoma [43], breast cancer [44], head and neck squamous carcinoma [45], laryngeal cancer [46], osteosarcoma [47], bladder cancer [48], glioma [49]
*CASC19*	Colorectal cancer [50], cervical cancer [51]
*PRNCR1*	Non-small cell lung cancer [52], tongue squamous cell carcinoma [53]
*CCAT1*	Colon cancer [54], endometrial cancer [55], prostate cancer [56], ovarian cancer [57], gastric cancer [58], hepatocellular carcinoma [59], breast cancer [60], cervical cancer [61], melanoma [62], laryngeal squamous cell carcinoma [63], glioma [64], oral squamous cell carcinoma [65], retinoblastoma [66], gallbladder cancer [67], oesophageal cancer [68], acute myeloid leukaemia [69], non-small cell lung cancer [70], osteosarcoma [71], nasopharynx cancer [72]
*CASC8*	Colorectal cancer [73], bladder cancer [74], lung cancer [75], hepatocellular carcinoma [76]
*CCAT2*	Colon cancer [77], pancreatic ductal adenocarcinoma [78], osteosarcoma [79], oesophageal carcinoma [80], hepatocellular carcinoma [76], acute myeloid leukaemia [81], non-small cell lung cancer [82], lung adenocarcinoma [83], ovarian carcinoma [84], endometrial cancer [85], breast cancer [86], renal cell carcinoma [87], epithelial ovarian cancer [88], glioma [89], prostate cancer [90]
*CASC11*	Colorectal cancer [91], cervical cancer [92], gastric cancer [93], bladder cancer [94], small cell lung cancer [95], non-small cell lung cancer [96], hepatocellular carcinoma [97], oesophageal carcinoma [98], neonatal neuroblastoma [99], ovarian squamous cell carcinoma [100], osteosarcoma [101], glioma [102]
*PVT1*	Non-small cell lung cancer [103], colon cancer [104], ovarian cancer [105], oesophageal adenocarcinoma [106], nasopharyngeal carcinoma [107], cervical cancer [108], glioma [109], bladder cancer [110], pancreatic cancer [111], hepatocellular carcinoma [112]
*TMEM75*	Colorectal cancer [113]
*CCDC26*	Acute myeloid leukaemia [114], acute monocytic leukaemia [115], chronic myeloid leukaemia [115], glioma [116]

**Table 2 ijms-22-01094-t002:** Summary of microRNAs for which lincRNAs in the 8q24.21 region act as a competitive endogenous RNA.

8q24.21 LncRNA	miRNAs and Affected Target Genes
*PVT1*	miR-424-5p (*CARM1*) [119], miR-145 (*FSCN1*) [124], miR-143 (*HK2*) [125], miR-128-3p (*GREM1)* [126], miR-186 (*Twist1*) [127], miR-216b (*Beclin-1*) [128], miR-30a (*IGF1*) [129], miR-20a-5p (*ULK1*) [130], miR-29c (*VEGF*) [103], miR-26b (*CTGF*) [131], miR-140-5p (*SMAD3*) [132], miR-150 (*HIG2*) [133], miR-365 (*ATG3*) [112], miR-17-5p (*PTEN*) [134], miR-31 (*CDK1*) [110], miR-128 (*VEGF*) [135], miR-1207-5p (*STAT6*) [136]
*CCAT1*	miR-181a-5p (*HOXA1*) [120], miR-143 (*PLK1*) [68], miR-148a (*PKC*) [56], miR-130a-3p (*SOX4*) [70], miR-218 (*ZFX*) [137], miR-148 (*PIK3IP1*) [71], miR-490-3p (*CDK1*) [138], miR-181a (*CPEB2*) [72], miR-218 (*ZFX*) [139]
*PCAT1*	miR-128 (*ZEB1*) [121], miR-145-5p (*FSCN1*) [140], miR-129-5p (*HMGB1*) [141], miR-122 (*WNT1*) [42]
*CCAT2*	miR-200b (*VEGF*) [122], miR-23b-5p (*FOXC1*) [142], miR-424 (*VEGFA*) [89]
*CASC11*	miR-302 (*CDK1*) [123], miR-676-3p (*NOL4L*) [99], miR-498 (*FOXK1*) [102]
*PRNCR1*	miR-944 (*HOXB5*) [53], miR-126-5p (*MTDH*) [52]
*CASC19*	miR-140-5p (*CEMIP*) [50]

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
