# Peer review of "8q24.21 Locus: A Paradigm to Link Non-Coding RNAs, Genome Polymorphisms and Cancer"

_ijms, 2021, doi:10.3390/ijms22031094_

Round 1
Reviewer 1 Report
The authors have reported an overview of genome polymorphisms of the 8q24.21 locus associated with cancer. Thus this report may be an interesting study. Especially, as it is located close to the c-myc proto-oncogene whose abnormal up-regulation leading to uncontrolled cellular proliferation has been well documented in many cancers.
Thus the importance of the 8q24.21 locus may be viewed as “grand”, albeit other unrelated lncRNAs have also been described in cancer.
For example, there are many similar reports about alterations in the lncRNA “Malate”, which is highly up-regulated in cancer. Therefore, functional assays concerning malate in mice are intriguing, such as the non-phenotype so far observed in a malate deleted mouse.
Thus the authors should at least mention mouse work if an equivalent locus to 8q24.21 exists in mus musculus or to mention as future plans to validate human data, or give their opinion about any expected phenotype.
Because polymorphisms in cancer cells are frequent, why should polymorphisms in the 8q24.21 locus be of greater importance? In addition, a number of genes are altered in cancer cells. Findings of up-regulation of a gene and /or lncRNAs in a given cancer has already been well documented. Thus experiments to get to their functionsl significance are lacking. Other than that this review provides a very complete view of alterations at the 8q24.21 locus.
Minor points:
References 1 and 2 are not in right format in the text.
Reviewer 2 Report
The review “8q24.21 locus: a paradigm to link non-coding RNAs, genome polymorphisms and cancer” provides a very interesting and detailed analysis of the relationships between lncRNAs, miRNAs and protein coding genes expressed by the 8q24.21 locus and their contribution to oncogenesis. The specific articles collected and presented in the text and tables are in general pertinent and up to date.
According to the most recent genome reference assembly, LRATD2 appears to be a protein coding gene, formerly known as FAM84B. Its contributions to oncogenesis have been investigated. Please revise Figure 2 and chapter 4 accordingly. Please also mention the other protein-coding genes that are present in the locus.
At line 146, please indicate whether CASC8 is sense or antisense to POU5F1B. For added clarity, it would be useful to represent with an arrow the direction of gene transcription in Figure 2.
In Table 1, please check ref. 50 and 86.
At lines 284, 287, please check ref. 54.
At lines 306, please check ref. 149.
